# A Low-Cost, Flexible Pressure Capacitor Sensor Using Polyurethane for Wireless Vehicle Detection

**DOI:** 10.3390/polym11081247

**Published:** 2019-07-27

**Authors:** Chien Khong Duc, Van-Phuc Hoang, Duy Tien Nguyen, Toan Thanh Dao

**Affiliations:** 1Faculty of Radio-Electronic Engineering, Le Quy Don Technical University, No. 236 Hoang Quoc Viet Street, Co Nhue 1 Ward, Bac Tu Liem District, Hanoi 100000, Vietnam; 2Faculty of Electrical-Electronic Engineering, University of Transport and Communications, No. 3 Cau Giay Street, Lang Thuong Ward, Dong Da District, Hanoi 100000, Vietnam; 3Faculty of Civil Engineering, University of Transport and Communications, No. 3, Cau Giay Street, Lang Thuong Ward, Dong Da District, Hanoi 100000, Vietnam

**Keywords:** flexible pressure sensor, polyurethane polymer, wireless data acquisition (DAQ), non-intrusive monitoring, vehicle detection

## Abstract

Detection of vehicles on the road can contribute to the establishment of an intelligent transportation management system to allow smooth transportation and the reduction of road accidents. Thus far, an efficient and low-cost polymer flexible pressure sensor for vehicle detection is lacking. This paper presents a flexible sensor for vehicle sensing and demonstrates a wireless system for monitoring vehicles on the road. A vehicle sensor was fabricated by sandwiching a polyurethane material between aluminum top/bottom electrodes. The sensing mechanism was based on changes in capacitance due to variation in the distance between the two electrodes at an applied external pressure. A clear response against a pressure load of 0.65 Mpa was observed, which is the same pressure as that of the car tire area in contact with the road. Significantly, the sensor was easy to embed on the road line due to its mechanical flexibility and large size. A field test was carried out by embedding the sensor on the road and crossing the sensor with a car. Moreover, the signal displayed on the tablet indicated that the sensing system can be used for wireless detection of the axle, speed, or weight of the vehicle on the road. The findings suggest that the flexible pressure sensor is a promising tool for use as a low-cost vehicle detector in future intelligent transportation management.

## 1. Introduction

Vehicle detection is an important process in intelligent transport management systems, as it allows the collection of big data on vehicles’ speeds and weights as well as the traffic intensity, which helps to enhance smooth transportation and to reduce road accidents [1]. Non-intrusive and intrusive sensor technologies are often employed for monitoring [1,2,3,4,5,6,7,8,9,10,11]. A laser sensor, a temperature sensor, or an image-based sensor have been used for vehicle detection that is based on a change in laser light intensity, temperature value, or imaging property due to the vehicle appearance [6,7,8]. These approaches have high potential to estimate traffic volume and vehicle classification. However, they are negatively affected by the environmental factors, for example fog, snow, rain, or shadow and the installation and maintenance of such sensor systems result in a relative high cost [8]. Recently, induction loops are most commonly used due to their low-cost and high reliability. The operating concept is that when a vehicle passes over the loop, it induces a current change in the wire loop [1,2,3]. The limitation of such sensors is that the installation process leads to significant damage to an extensive area of road surface. A recent study employed anisotropic magnetoresistance sensors [9,10,11]. Determination of vehicles was realized because the metal construction of a vehicle distorts the magnetic field of the Earth. However, the drawback of this sensing method is that it has a relatively high cost, and is likely to contain noise signals due to interference from neighboring vehicles on the road or metal materials from other sources.

On the other hand, in recent years, a pressure sensor using polymer materials has received much attention among academics and industry due to its unique advantages, including low-temperature processing, low manufacturing costs, mechanical flexibility, and the potential to work in a large area [7,8,12,13,14,15,16,17,18,19,20,21,22,23]. These characteristics indicate the potential to construct the next generation of IoT (Internet of Things) sensing nodes, where the sensor is required to not only be low-power and low-cost, but also, be compatible with many surface types [11,12]. Structurally, the sensor contains an active polymer material layer sandwiched between two electrodes [8,12,13,14,15,16,17,18,19,20,21,22,23]. Based on the physical phenomena of piezoresistivity, piezocapacity, and piezoelectricity, when the applied pressure changes, the resistance, capacitance, or electricity of the sandwiched layer varies, which, in turn, causes the output electrical signal of the device to change. In general, those sandwiched layers are mostly obtained through the use of sensing material system such as a poly(vinylidenefluoride-co-trifluoroethylene (P(VDF-TrFE)), a polydimethylsiloxane (PDMS) or a mixture of polymer and nanoparticles or polymers doped with carbon nanotube [8,12,13,14,15,16,17,18,19,20,21,22,23]. Based on its advantages, the flexible pressure sensor is expected to be used in vehicle detection without damaging the transportation infrastructure, as it can be adhered to the road. In addition, the sensor may not be affected by the magnetic field of Earth like an anisotropic magnetoresistance device since the polymer is a non-metallic material. In particular, literature studies have mainly focused on the use of flexible sensors with a pressure from a low-regime (0–0.01 Mpa) to medium-regime (0.01–0.1 Mpa) for human health monitoring, such as heart rate monitors, EEG signals, object tracing, collision events, or weight measurements [11,12,13,14,15,16,17,18,19,20]. In addition, a sensor with a higher measurement range has recently attracted a great deal of attention because of its promising applications in smart transportation [7,8,24]. For example, Ding et al. demonstrated a sensor device by adding spiky spherical nickel particles into liquid silicon rubber. In such nickel particles-based sensor, the sensed value was in the range of 0–6.4 Mpa [7]. Mohiuddin and Van Ho fabricated a pressure sensor by mixing the multiwalled carbon nanotubes in a polyether ether ketone polymer matrix, which resulted in the measurement value up to 40 Mpa [24]. Overall, those sensors are capable for utilization in vehicle sensing because their measurement ranges are well overlapped the pressure of car tires on the road (0.4–0.7 Mpa, Ref. [25]). However, the device fabrication technique including the control of uniform dispersion of dopants in the polymer matrix is relatively complicated and costly, which limits the wider use of sensors to practical public transportation.

In this work, we demonstrate a vehicle detection capacitor sensor fabricated with a polyurethane material and a thermal lamination technique. We find that by applying an external force to the sensor, the thickness of the polyurethane dielectric layer is decreased, resulting in increased capacitance of the sensor. The measurement value of the sensor is up to 0.65 Mpa, which matches the pressure of a vehicle tire well [25,26]. The best sensitivity of the sensor is obtained to be 8 × 10^−2^ kPa^−1^. The sensor is easy to mount on a road without re-construction of the road surface due to its flexibility. Additionally, wireless data acquisition (DAQ) together with a tablet computer-based application (App) is also developed for vehicle detection. Real-time vehicle sensing is successfully achieved by the sensor with the wireless DAQ system embedded on the road and the tablet computer.

## 2. Experimental Method

Figure 1 shows a graphical illustration of the flexible sensor where a capacitor structure was utilized because of its simple structure, good stability, and low power consumption [3,9,10,11,12]. For sensor fabrication, aluminum (Al) foil electrodes purchased from Sigma-Aldrich (resistivity of 2.6548 μΩcm; thickness of 30 μm, Singapore) were initially sputter-etched to remove potential contamination or native oxide structures. In short, the Al foil was loaded in the etching chamber, under a vacuum background of 10^−6^ Torr. Subsequently, argon gas with a flow rate of 20 sccm is inlet into the chamber, and the etching pressure is controlled to be 0.1 Pa. An automatic tuning process is applied to generate plasma atmosphere, and to start an etching process. The etching power is set to be 30 W, corresponding to the etching rate of 0.5 nm/min. For our case, we performed this cleaning process about 5–10 min to ensure a perfect conductive surface of Al foil. A 100-μm-thick polymer film of polyurethane (Tensile strength = 38 Mpa, Sanyo Chemical, Tokyo, Japan) was employed as an active layer, which was sandwiched between two aluminum electrodes, followed by lamination at 80 °C twice (Figure 2a). The sensor size was determined by the overlapping area of the two electrodes to be 70 mm × 70 mm. This large size was used with the aim of being suitable for an application in car tire tracking. Subsequently, a copper wire was attached to the electrode to form the connecting lead of the sensor. Finally, for protection, the sensor was also covered with a plastic film by the lamination method at 80 °C (Figure 2b). The rollers of the laminator in here are to help to flatten sandwiched polymer and increase the adhesive functionality. Figure 2c shows photos of the fabricated sensor in a (left) normal or (right) flexible mode. To examine the homogeneity of sensor layer after lamination, a cross-sectional image of the sensor was taken with a microscope. As can be seen in Figure 3, each layer is well realized, indicating a homogenous property of device layer.

Characterization of the sensors was performed using a universal compression testing machine (UH 500-kN, SHIMADZU, Kyoto, Japan) and a capacitance meter (YF-150, Tenmars Electronics, Taipei, Taiwan) at a frequency of 100 Hz at room temperature. The dynamic range of the sensor was tuned to vary from 0 to around 0.65 Mpa, which is the same pressure as the car tire area contacts with the road. To examine the influence of the polymer thickness in sensing, devices with different film thicknesses of polyurethane (200, 300, and 500 μm) were also fabricated using the same experimental method.

## 3. Results and Discussion

### 3.1. Sensor Characteristics

Figure 4 shows the capacitance-pressure characteristics of the sensors when the pressure varies from 0 to 0.65 Mpa. The initial capacitance values were measured to be 0.13, 0.19, 0.083, and 0.050 pF/mm^2^ from the devices with polyurethane thicknesses of 100, 200, 300, and 500 μm, respectively. A steep increment in capacitance was observed at the low-pressure region and the increment tended to saturate with an increasing applied pressure (*p*). A similar characteristic is also typical among capacitive sensor systems [17,19,20,21], suggesting that our device sensors were well fabricated.

To understand the operating mechanism of the fabricated sensors, we carried out further analysis, as follows. Fundamentally, the capacitance of two parallel electrodes can be described by the below equation:(1)$$C=\varepsilon \frac{A}{t}$$
where *ε*, *A*, and *t* are the dielectric constant, the area of overlap of the two electrodes, and the thickness of the dielectric layer (or the distance between the electrodes), respectively. As shown by Equation (1), under application of pressure, the change in *C* can be considered for the following reasons: (1) a change in the dielectric constant of the dielectric layer, (2) a change in the device area, or (3) a change in the distance between two electrodes. However, mechanism (1) can be neglected because the dielectric constant of the polymer material obtained during the sensor under compression was almost unchanged to be about 2.2. If the device area change is the reason, the capacitance has difficulty rapidly increasing like in curve Figure 4. Thus, we assume that the decrease in the thickness thanks to the elasticity of the polymer is attributed to the operating mechanism. Based on that consideration, the proposed operation mechanism of the sensor is presented in Figure 5. Under application of force to the sensor electrode, the thickness of the film decreases, which, in turn, results in the capacitance increasing in accordance with Equation (1). In contrast, the applied force is removed from the interface; consequently, the capacitance reverts to the initial value.

We conducted an estimation to investigate the variation in *t* with external pressure loading. From Equation (1), *t* was calculated using the following equation: *t* = *εA*/*C*. For example, in Figure 6, the capacitance change and the thickness change with respect to the external pressure are plotted for the 100 μm thick film sensor. The thickness *t* of the layer tends to decrease with increasing pressure which is the opposite tendency to the chance in capacitance.

In sensors for human health monitoring [16,17,18,19,20,21,22,23], the void structures have widely used to enhance sensitivity of the sensors, but those also result in the measurement value of below 0.1 Mpa. In our study, thanks to the utilization of a polyurethane film with a tensile strength of 38 Mpa, the ability of the sensor to measure pressure values of up to 0.65 Mpa.

On the other hand, in order to quantify the influence of the active thickness, the effective capacitance change was estimated with the following equation [21]:(2)$$\frac{\Delta C}{C_{0}}=\frac{C_{\max}-C_{0}}{C_{0}}$$
where *C*_max_ is the capacitance at *p* = 0.65 Mpa, and *C*_0_ is the capacitance at *p* = 0. We found that the Δ*C*/*C*_0_ significantly decreased from 1.070 to 0.274, 0.172 and 0.116 when the polyurethane thickness increased from 100 to 200, 300 and 500 μm, respectively. This could be due to the fact that the reduction in the distance between the top and bottom electrodes in a thin device is more considerable.

The sensitivity (*S*) was calculated using the equation [21]:(3)$$S=\frac{\delta (\Delta C/C_{0})}{\delta P}$$

The *S* at three pressure ranges and different film thicknesses is shown in Table 1. The thinner device exhibits a higher *S*, which results from its factor of Δ*C*/*C*_0_ being higher. The *S* values in the range of 0.2–0.65 Mpa for each device are 1.18 × 10^−4^, 5.57 × 10^−5^, 5.07 × 10^−5^, and 8.35 ×10^−5^, respectively. Overall, the 100 μm-based sensor exhibits the best performance in terms of Δ*C*/*C*_0_ and *S*. Significantly, an *S* value of up to 8 × 10^−2^ kPa^−1^ was obtained from the 100 μm based-sensor at 0.003 Mpa, which is comparable to the previous pressure sensor systems [19,20,21].

In order to confirm the reproducibility of the change in capacitance values under constant *p*, the repeatable properties of the sensor in response to a constant *p* of 0.65 Mpa were checked with the 100 μm based-sensor (Figure 7). In the first cycle, capacitance at *p* = 0 Mpa was not returned to the initial value when the pressure was released. This would be due to the deformation of the polyurethane film. However, in the second cycle, the repeatable characteristics were well observed as shown in Figure 7a. Moreover, a pressure-sensing characteristics under various values of *p* were investigated, wherein the *p* values of 0.08, 0.2, 0.65, 1.0, and 1.5 Mpa were applied. As can be seen in Figure 7b, the sensor presents stable responses with a great repeatability. Overall, the sensors can be considered durable for the pressure and available for vehicle sensing.

Figure 8 presents flexibility of polyurethane sensor where the capacitance of the sensor was measured at different bent radii of curvature ranging from 200 mm to infinite (normal state). As can be seen in Figure 8, a relative steep increment in capacitance was observed at the bent radius from 200 to 500 mm and the increment tended to saturate at bent radius above 500 mm, suggesting that the polyurethane sensor can be properly operated at bent radius more than 500 mm. We would like to confirm here that, the ability of flexible operation makes the sensor adapt well to the surface roughness of the asphalt concrete (see Appendix A) which is widely used to construct a road nowadays.

### 3.2. Development of Wireless DAQ

For the purpose of the field test using the flexible pressure sensor, a completed electronic system using the sensor and a wireless DAQ was designed with the circuit diagram presented in Figure 9, where a NE555 based-oscillator [27,28] was utilized to convert the sensing capacitance to a pulse train and then digitized with a 10-bit-ADC (analog-to-digital converter). With the aim of constructing a DAQ with a small size and low-power, an IoT microprocessor of L106 32-bit RISC and IEEE 802.11 b/g/n integrated in the ESP8266 module were used. An embedded code for the microprocessor chip was written to process, store, and transmit the sensed data to the tablet computer through a WiFi method. The power supply for the DAQ circuits in DAQ was provided from a 3.7 V-LiPo rechargeable battery. A tablet app written with Android Studio allowed real-time display of the sensed signal and export of the data in Excel format for further analysis. We note that since capacitance might not be returned to the initial value when the pressure was released after the first cycle as shown in Figure 7a, the signal data were normalized before displaying on Tablet.

In order to test durable operation for sensor device system consisting of the flexible sensor, wireless DAQ and Tablet app, a Wheel-Track Device of Hamburg AASHTO T 324-04 with a loaded pressure of 0.6 Mpa was used. As shown in Appendix A, our sensor device system can be well operated for several hundreds of cycles without degradation, suggesting a highly stable sensor device system.

### 3.3. Demonstration of Vehicle Sensing

Figure 10 shows a University field test of the vehicle sensing using the system where the sensor was easy to embed between the paper hardcover and was mounted on the road line using tape without any re-construction of the road surface. We would like to note here that our system is advanced compared to current vehicle sensing technology where the road must be re-constructed [1,9]. The sensor is easily mounted on the road due to its flexibility. The relative large sensor size of 70 mm × 70 mm can make it convenient for matching the footprint of the tire. In this demonstration, a Honda car of 1000–1400 kg with two axles was used as the test vehicle. The signal data for each test were real-time recorded and stored, for example, Appendix A show the recorded signal when the car is passing at 0, 5, and 20 km/h, respectively. Figure 11 shows a typical reconstructed signal of axle load. Based on that, the axle, speed, or weight of the vehicle was deduced.

The primary goal of the sensor device system is to detect the vehicle axle, leading to a purpose of vehicle classification. Figure 11 shows a typical axle signal reconstructed during a vehicle passing on the sensor (see Appendix A). It is well-known that a peak impulse in signal output corresponds to a vehicle axle [8,29,30]. As can be seen, the 2-pulse signal presents accurately a 2-axle car.

The vehicle speed (*V*) was estimated with the following equation [29]:(4)$$V=\frac{d}{t}$$
where *d* is the distance between axles of vehicle (Wheelbase) and *t* is the period between the first and second pulses (Figure 11).

The vehicle weight (W) was estimated with the following equation [29]:(5)$$W=V\times A\times K_{W}$$
where *A* is the area of pulse (Figure 11) and *K*_W_ is the calibration factor, which is obtained by using a passing vehicle with known weight and sensor width.

Table 2 lists the true *V* and calculated *V* based on sensor output data for a Honda car with a known *d* of 2600 mm. As shown, the error increases as increasing the true *V*. Overall, the error value is relative high, for example, at true *V* of 30 Km/s, the error is estimated to be more than 10%. Thus it needs to further study on the algorithm in order to increase the accuracy of the vehicle speed for our sensor system. At the true *V* of 35 km/h, a pulse signal cannot be observed. This is because the smaller deformation for the sensor at higher speed, leading to the area of pulse signal becomes narrower as the speed increases. Thus, it can be concluded that the *V* detection of current sensor device is less than 35 km/h. On the other aspect, Table 3 shows results of estimated *W* and errors at *V* = 5 km/h when the true *W* varied from 1000 to 1400 kg. As shown in Table 3, the calculation of the *W* is more accurate with an error less than 0.5%. It notes here that the range of true *W* is narrow because our limitation of weight source facility. However, based on the first results from the estimations of *V* and *W*, it indicates that the sensor system is capable of determining axle, collecting the data of *V* or *W* and is useable for vehicle detection.

## 4. Conclusions

In conclusion, a flexible sensor for vehicle detection made was fabricated with polyurethane material using a lamination method. The operating mechanism was attributed to the reduction of the thickness of the dielectric layer *t*. Electrical characterization revealed that the 100 μm-based sensor exhibits the best performance in terms of Δ*C*/*C*_0_ and *S*. The ability of the sensor to measure pressure values of up to 0.65 Mpa shows that it has potential for application in vehicle detection. Moreover, a wireless system based on the fabricated sensor for monitoring vehicles on the road was demonstrated. Owing to its mechanical flexibility and large size, the sensor is easy to embed on the road line. The axle, speed, or weight of the vehicle can be realized based on the app on the tablet computer. The experimental results presented here indicate that the flexible pressure sensor is a promising way to construct a low-cost intelligent transportation management device. However, obviously, the sensor systems have still some problems such as a low-accuracy of the estimated *V* and low measurement value (less than 35 Km/h), suggesting that additional works especially in algorithm need to be performed in order to suppress those limitations. In addition to that, more tested scenarios including utilizations of a wider weight range, real road, or different types of vehicle are necessary to be done to evaluate for further validation of the fabricated sensor device systems.

## Figures and Tables

**Figure 1 polymers-11-01247-f001:**
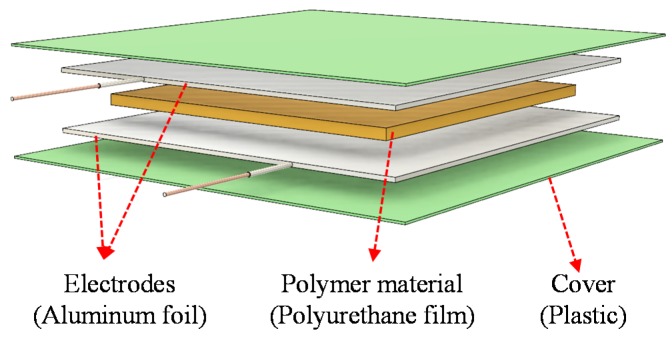
Structure of the flexible pressure.

**Figure 2 polymers-11-01247-f002:**
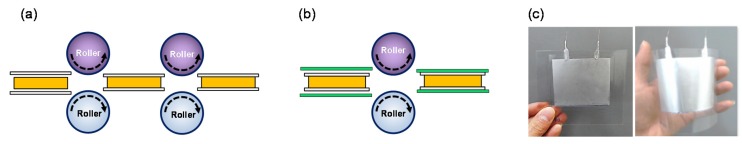
Illustration of the sensor fabrication process: (**a**) electrode layer formation, (**b**) encapsulation, (**c**) photos of the sensor in (left) normal or (right) flexible mode.

**Figure 3 polymers-11-01247-f003:**
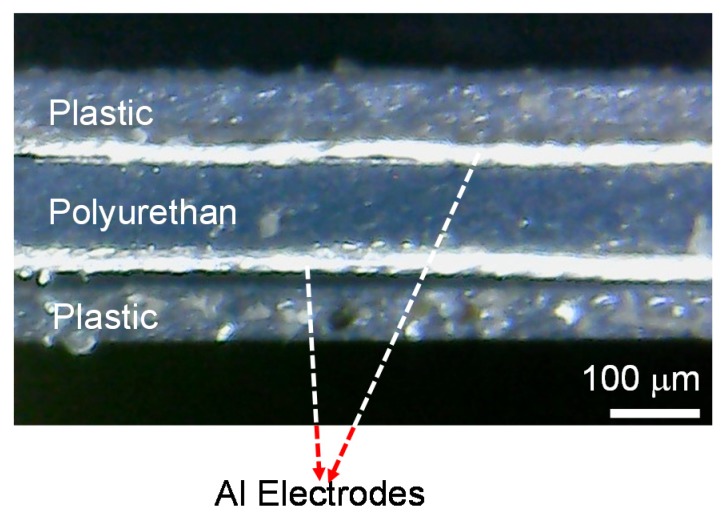
Cross-sectional image of fabricated sensor.

**Figure 4 polymers-11-01247-f004:**
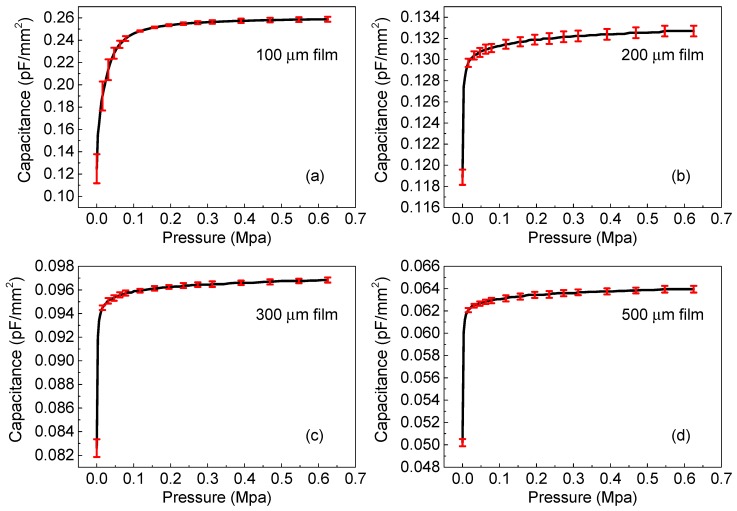
Capacitance-pressure characteristics of sensors with thicknesses of (**a**) 100, (**b**) 200, (**c**) 300, and (**d**) 500 μm.

**Figure 5 polymers-11-01247-f005:**
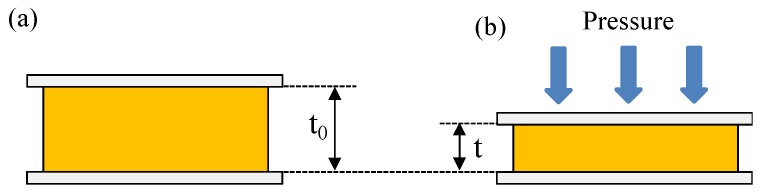
Illustration of the operating mechanism in pressure capacitor sensor: (**a**) without pressure and (**b**) under pressure.

**Figure 6 polymers-11-01247-f006:**
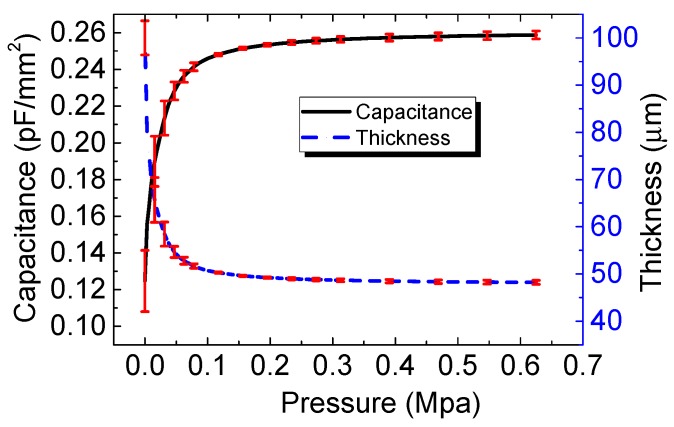
Capacitance and the thickness change with respect to the external pressure.

**Figure 7 polymers-11-01247-f007:**
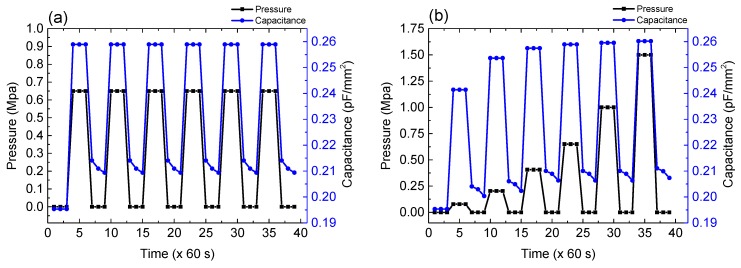
Endurance cycle test of the 100 μm-based sensor in response to (**a**) constant *p* and (**b**) various values of *p*.

**Figure 8 polymers-11-01247-f008:**
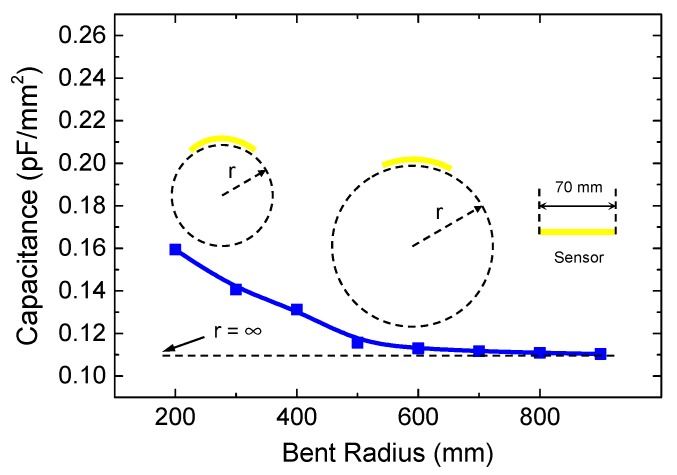
Measured capacitance of capacitor sensor at different bent radius. Inset shows illustration of flexible operation at different bent radius.

**Figure 9 polymers-11-01247-f009:**
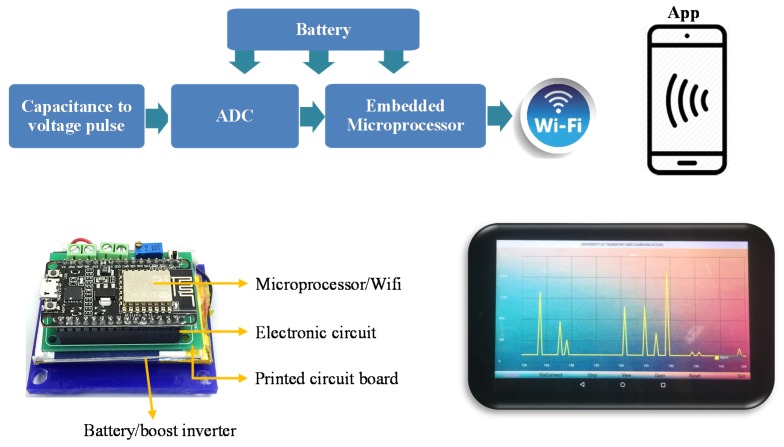
Diagram of the wireless data acquisition (DAQ) device. Inset shows the printed circuit board (PCB) and tablet with the running app.

**Figure 10 polymers-11-01247-f010:**
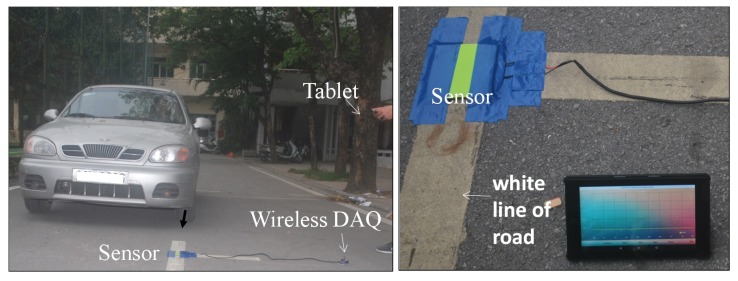
(**left**) Experimental overview of the vehicle sensing using the system and (**right**) close-up view of sensor mounted on the road.

**Figure 11 polymers-11-01247-f011:**
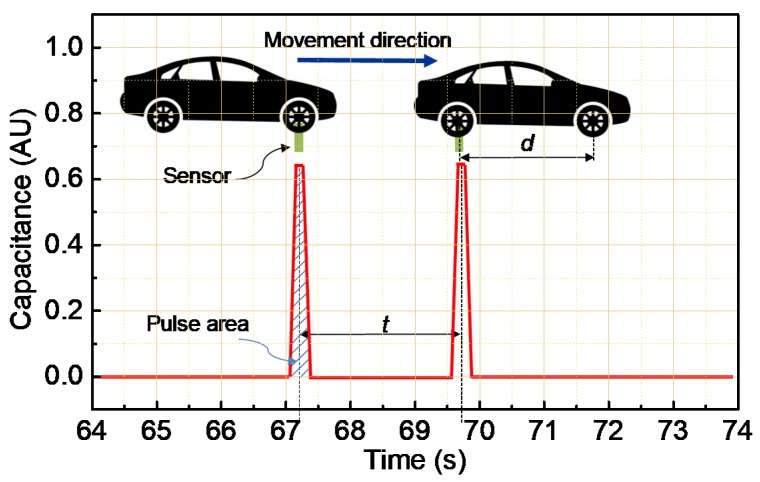
Reconstructed signal of axle load recorded at 5 km/h. Inset illustrates related parameters to estimate speed or weight vehicle.

**Table 1 polymers-11-01247-t001:** Sensitivity at different pressure ranges or film thicknesses.

Pressure Range (Mpa)	Sensitivity (kPa^−1^)			
	t = 100 μm	t = 200 μm	t = 300 μm	t = 500 μm
0–0.03	2.42 × 10^−2^	3.40 × 10^−3^	5.33 × 10^−3^	8.77 × 10^−3^
0.03–0.2	2.98 × 10^−3^	1.64 × 10^−4^	1.89 × 10^−4^	2.30 × 10^−4^
0.20–0.65	1.18 × 10^−4^	5.57 × 10^−5^	5.07 × 10^−5^	8.35 × 10^−5^

**Table 2 polymers-11-01247-t002:** Results of estimated *V* and errors for the different true *V* values.

No.	True V (km/h)	Estimated V (Km/h)	Error (%)
1	2	2.02	1.00
2	5	5.13	2.60
3	10	10.35	3.50
4	15	15.61	4.06
5	20	21.21	6.05
6	25	26.79	7.16
7	30	33.25	10.83
8	35	N/A	N/A

**Table 3 polymers-11-01247-t003:** Results of estimated *W* and errors for the different true *W* values at *V* = 5 km/h.

No.	True W (Kg)	Estimated W (Kg)	Error (%)
1	1000	999.51	0.049
2	1050	1048.45	0.15
3	1100	1104.05	0.37
4	1200	1205.46	0.46
5	1300	1296.15	0.30
6	1400	1404.56	0.33

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
