# Peer review of "A Low-Cost, Flexible Pressure Capacitor Sensor Using Polyurethane for Wireless Vehicle Detection"

_polymers, 2019, doi:10.3390/polym11081247_

Round 1

Reviewer 1 Report

For figure 8 - How bend radius is related to the sensor performance? it should be explained and the text related to the figure is ambiguous. Properly operated with more than 500 mm bend radius? but with more than 500 mm, capacitance is less. do you want less capacitance? 

Figure 1 - Are electrodes, polymer and the cover are in ratio of their actual thickness? If not, then " Not to Scale" should be written. Then you dont need to add 20 mm scale bar.  

Reviewer 2 Report

A simple, yet functional sensor has been presented for detecting cars on the road and on intersections. The sensor is based on capacitance variations that occur when the car’s wheel exerts pressure on the parallel plates of the sensor. The manuscript has been extensively reviewed from its original version, but still there are some corrections to be done before acceptance and publication, they are next listed:

Line 66-71: I agree with the authors in that most sensing applications report nominal pressures of tens or hundreds KPa at most; this is so because the development of wearable devices is a hot trend nowadays. However, there are also multiple studies that have characterized force/pressure sensors within the range of tens MPa [1]. So please, reconsider your statement in Line 71.

Figure 7 provides important information to the reader; the figure reports that the sensor can detect cars during loading-unloading cycles. However, information is missing about how long the sensor takes to fully recover between subsequent loadings, i.e. how long is required for the sensor’s capacitance to show the null-stress reading after a full load? Besides this, could you specify if the sensor loses repeatability after several trials (100 trials or ever more).

References

[1] doi: 10.1186/1556-276X-6-419

Round 2

Reviewer 2 Report

Suggested changes and concerns have been performed. The article can be published in present form.